# Insights into AGB Nucleosynthesis Thanks to Spectroscopic Abundance Measurements in Intrinsic and Extrinsic Stars

Sophie Van Eck [1,*] , Shreeya Shetye [2] and Lionel Siess [1]

1   Institut d'Astronomie et d'Astrophysique, Université libre de Bruxelles (ULB), CP 226,
    B-1050 Bruxelles, Belgium; lsiess@ulb.ac.be
2   Laboratory of Astrophysics, Institute of Physics, Ecole Polytechnique Fédérale de Lausane (EPFL),
    Observatoire de Sauverny, 1290 Versoix, Switzerland; shreeya.shetye@epfl.ch
*   Correspondence: svaneck@astro.ulb.ac.be; Tel.: +32-2-650-28-63

**Abstract:** The foundations of stellar nucleosynthesis have been established more than 70 years ago. Since then, much progress has been made, both on the theoretical side, with stellar evolution and nucleosynthesis models of increasing complexity, using more and more accurate nuclear data, and on the observational side, with the number of analyzed stars growing tremendously. In between, the complex machinery of abundance determination has been refined, taking into account model atmospheres of non-solar chemical composition, three-dimensional, non-LTE (non-local thermodynamic equilibrium) effects, and a growing number of atomic and molecular data. Neutron-capture nucleosynthesis processes, and in particular the s-process, have been scrutinized in various types of evolved stars, among which asymptotic giant branch stars, carbon-enhanced metal-poor stars and post-AGB stars. We review here some of the successes of the comparison between models and abundance measurements of heavy elements in stars, including in binaries, and outline some remaining unexplained features.

**Keywords:** asymptotic giant branch (AGB); nucleosynthesis; s-process; intrinsic stars; extrinsic stars

## 1. Introduction

Clues to constrain nucleosynthetic processes come from many different stellar objects. Historically, it is the observation, 70 years ago, of technetium in the photosphere of a star (the S-type star R And, [1,2]) which put an end to the controversy between the proponents of primordial (G. Gamow) and stellar (F. Hoyle) nucleosynthesis, since it made it clear that stars could produce heavy chemical elements. Since then, many observations, followed by careful analyses consisting in chemical element detections or quantitative abundance determinations, have advanced our current understanding of element nucleosynthesis in stars. The level of accuracy of abundance determinations, together with the quantity and quality of analysed objects, allows more thorough comparisons to be performed with theoretical predictions. In particular, Thermally-Pulsing Asymptotic Giant Branch (TPAGB) nucleosynthesis has benefited from major theoretical advances (e.g., [3–6] and references therein). During this phase of evolution, favourable conditions for the operation of the slow neutron-capture process (s-process), which is responsible for the production of half of the elements heavier than iron, are encountered. This nucleosynthetic process is characterized by the fact that the neutron capture timescales are in general slower than $\beta$-decay rates, and therefore produces isotopes located along the valley of $\beta$-stability; it takes place at neutron densities in the range of $10^6$–$10^{11}$ cm$^{-3}$ (e.g., [7]). Two main astrophysical sites for the s-process have been identified: AGB stars and massive stars (during the core He burning and in the convective carbon-burning shell of stars with $M > 8 - 10 M_\odot$ [6]). In the following, only the observationally well-constrained s-process in AGB stars will be considered. Two possible neutron sources have been identified: $^{13}$C($\alpha$,n)$^{16}$O, predicted to

be efficient in the H-He intershell and $^{22}$Ne($\alpha$,n)$^{25}$Mg, requiring higher temperatures and operating in AGB stars more massive than 4 M$_\odot$.

Observational targets providing constraints on AGB nucleosynthesis can be classified in two types: intrinsic and extrinsic stars. Intrinsic stars experience or have experienced nucleosynthesis and brought to their photosphere the synthetized elements. Extrinsic stars have been polluted by a companion AGB star but did not (yet) reach the TPAGB stage. Finally, chemical enrichment of the Galaxy, resulting from a collection of pollution episodes, not accross a binary system like for extrinsic stars, but within the interstellar medium, can also put extremely useful constraints on AGB nucleosynthesis.

Evidences for a scenario of mass transfer between two stars are manyfold: barium stars are binaries [8,9], as well as no-Tc S stars [10] and unevolved objects of various metallicities showing carbon and s-process enhancement (CH stars and Carbon-Enriched Metal-Poor (CEMP) stars like CEMP-s stars and CEMP-sr stars [11]).

Here we describe some (among many) diagnostics used to constrain AGB nucleosynthesis. In the first section we review technetium detection in various types of stars, since technetium is the only unambiguous tracer, available from optical spectra, of on-going nucleosynthesis. In the second section we detail some comparisons between theoretical predictions and s-process element abundance determinations. Finally, we present some existing tensions between stellar evolution or nucleosynthesis models and the current abundance measurements.

## 2. Technetium, Intrinsic Stars and Extrinsic Stars

Technetium does not have any stable isotope, but has three long-lived ($^{97}$Tc, $^{98}$Tc and $^{99}$Tc) and 20 short-lived isotopes and isomers. $^{99}$Tc is the isotope produced by the s-process and it is the one assumed to be observed in stellar spectra. Since its lifetime ($\tau = 2.1 \times 10^5$ yr) is roughly one order of magnitudes shorter than the predicted lifetime on the AGB, any observed Tc must have been formed in situ. The 3 resonance lines of Tc I (4238.191, 4262.270, 4297.058 Å) are detectable, though heavily blended in spectra of cool stars.

Ionized technetium lines (Tc II) have also been searched in the spectra of barium stars and of low-metallicity AGB stars (they follow evolutionary tracks bluer and warmer than their solar-metallicity counterparts, so Tc, if present, could be partly ionized), without success [12,13] thereby confirming the extrinsic nature of barium stars. Tc II has also been searched in the spectra of some Ap stars, since it has been claimed that (some of) their chemical peculiarities (in particular concerning actinides) could be explained by nucleosynthesis processes involving particles accelerated by magnetic fields. These studies (e.g., [13]) lead to discrepant and inconclusive results, because they were essentially based on line coincidence identifications in the absence of adequate (non homogeneous) model atmospheres.

Finally, molecular lines of TcO have been looked for in (intrinsic) S-type stars without success [14].

Little-Marenin and Little [15,16] achieved Tc I line detection for 279 late-type giants of the M, MS, S and C spectral types, complemented since then by [17–20]; whether these technetium detections agree with stellar evolution models is discussed in the next paragraphs.

### 2.1. S-Type Stars

The least luminous AGB stars to exhibit intrinsic nucleosynthesis products are S-type stars, that display ZrO bands in addition to the TiO bands characterizing M-type stars. Combining the spectra analysis with Gaia parallaxes produces a consistent picture. For example, a nice agreement between the STAREVOL evolutionary models [21,22] and observations [19,20,23] is illustrated in Figure 1: as expected, Tc-rich stars are located above the threshold luminosity ($L_{3DU}$) corresponding to the onset of the third dredge-up of the matching metallicity [1] while no-Tc stars have a luminosity below $L_{3DU}$. Two stars are however discrepant in Figure 1: HD 150922 and BD-10° 1977, both no-Tc stars (squares) in the [Fe/H] = −0.5 bin (in green) falling above and to the right of the $L_{3DU}$ line. Whether

their binarity could have affected their Gaia eDR3 parallax remains to be checked with the next Gaia data release. With the exception of these two no-Tc stars, all Tc-rich stars fall above and to the right of the $L_{3DU}$ line of the corresponding metallicity, as expected (see also [19,20,24]).

The very large majority of no-Tc stars has been found to be binaries [25–28] and thus extrinsic stars. The proportion of no-Tc stars among S stars, initially estimated around 30% (and 26% in carbon stars [29]), has been revised to ~50% [10].

Thanks to MARCS [30] model atmospheres of the adequate, non-solar, chemical composition, it has been possible not only to assess the presence or absence of Tc in S-type stars, but also to derive Tc abundances. These abundances are subject to large uncertainties, mainly because the agreement between observed and synthetic spectra at the location of the three Tc resonance lines, around 4250 Å, is not satisfactory. However there is a loose tendency for higher Tc abundances among S stars with larger [s/Fe] and C/O ratios [20]. As far as S stars are concerned, the picture is thus roughly consistent with evolutionary models (more evolved AGB stars have experienced more dredge-ups so the Tc enrichment increases with increasing C and s abundances) but, as we shall see, remains fuzzy for MS and M stars at the low C/O limit, and for SC stars at C/O close to 1.

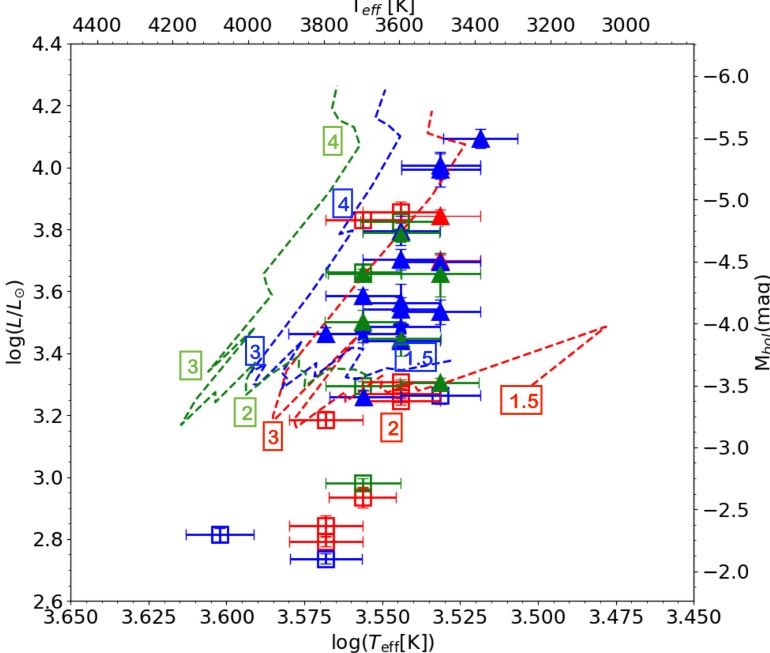

**Figure 1.** Hertzsprung-Russell diagram of S-type stars, using Gaia eDR3 parallaxes, from the samples of [19,20,24]. Intrinsic, Tc-rich S stars are represented with filled triangles and extrinsic, no-Tc stars by open squares. The colours separate stars in three different metallicity bins centered on [Fe/H] = −0.5 (green), −0.25 (blue) and 0 (red). The dashed lines connect the luminosity marking the onset of the third dredge-up events ($L_{3DU}$) for stars of masses between 1 and 5 $M_\odot$, according to the STAREVOL evolutionary tracks [21,22], with the same metallicity colour coding as for the stars. The masses of the stars experiencing this first 3DU are indicated in the colour squares (see Figure 4 of [20] for more details). Figure adapted from [19,20,24].

### 2.2. Technetium-Rich M Stars

Some stars, despite being classified as M (implying a very low or nul s-process enrichment), have been found to be Tc-rich. The question of technetium among M-type stars is difficult to settle, because there is a large proportion of variable stars, including Mira stars (variable stars with *V*-band photometric amplitude exceeding 2.5 mag) in this category, further complicating abundance determinations with static model atmospheres. M stars with technetium have spectral types later than M2 [16], and, counter-intuitively enough,

Mira stars with no-Tc show redder $K - [22]$ colours than Tc-rich Miras of similar pulsation periods, suggesting a higher dust content in their circumstellar environment [31]. Whether third dredge-up events would decrease the dust contribution remains to be proven. Actually it is not even clear whether Tc-rich M stars have only experienced a (few) thermal pulse(s) (so that the star would switch from the "no-Tc" to the "Tc" category, without increasing significantly its Zr abundance above the solar-scaled value), or whether they are the site of a different version of the s-process (producing Tc with negligible amounts of Zr [32]). Changes in average neutron-capture cross sections can also affect the comparison between predicted and measured abundances (e.g., [33] for technetium).

### 2.3. Nitrogen-Rich, No-Tc M Stars

A group of stars of still unclear status is the so-called "Stephenson M-type stars" uncovered in [34] [2]. These M-type stars have a large N abundance (typically $[N/M] \geq +1$, where M denotes iron-peak elements) but no carbon nor s-process enrichment. They were tentatively explained as TPAGB stars in an early stage of third dredge-up or interloping supergiants that have experienced the second dredge-up. They could be linked to the LMC post-AGB star J005252.87-722842.9 and its two potential galactic analogs, presenting no carbon nor s-process enrichment [36] (though the metallicity range is slightly different: $[-0.5; 0]$ for Stephenson M stars, close to $-1$ for the post-AGB stars of [36]).

### 2.4. MS Stars

The situation for MS stars is quite clear, since already [16] noted that their so-called "evolutionary" (i.e., intrinsic) MS stars showed Tc while "spectroscopic" (i.e., extrinsic) MS stars did not. The classification is thus nicely consistent with that of S stars. However, contrarily to S-type stars, the binarity of no-Tc MS stars has not yet been established.

### 2.5. SC Stars

Concerning SC stars (the distinction between SC and CS stars will not be considered here), characterized by a C/O ratio of quite exactly [3] one, we are aware of only 3 stars where the presence of Tc could be conclusively assessed (R CMi, CY Cyg, RZ Peg [16]). Upper limits for Tc abundance measurements are also available for 6 SC stars [17]. Again, this is consistent with an S–SC–C evolutionary sequence on the TPAGB, as the dredge-ups increase the C/O ratio from 0.5 to 1 (SC stars) and then to values larger than 1 (C stars).

### 2.6. C Stars

Technetium measurements are especially complicated in carbon stars due to the strong lack of flux at 4250 Å, in the region where the Tc I resonance lines are located, so that Tc detection is rather based on the pair of accessible intercombination lines (5924.468 and 6085.229 Å) resulting from semi-forbidden transitions [29]. Following [37] we distinguish C-J (or J-type) stars (solar-metallicity carbon stars with a low $^{12}C/C^{13} < 15$ ratio, with normal N abundance but often Li-rich), C-R or R-type (warm carbon stars) and C-N or N-type (cool carbon stars). Tc has been detected in most C-N stars where it was searched for, but not in the few investigated J stars (like Y CVn), which do not appear to be enriched in s-process elements either [16,38,39]. Finally, R stars apparently do not show s-process enhancements [40] and have lower luminosities ($M_{bol}$ mostly $> -1$) than the J, SC and N stars ($-4 < M_{bol} < -6$) [41], the latter being in agreement with TPAGB luminosities (see Figure 1). The Tc detection attempts in R stars remain unconclusive [40].

## 3. The s-process along the AGB

The s-process has been validated in several types of TPAGB objects. Observations allow one to constrain in particular the neutron source ($^{13}C(\alpha,n)^{16}O$ or $^{22}Ne(\alpha,n)^{25}Mg$), the neutron density (that can be inferred through various branching points in the s-process path, e.g., [42]), and the mixing processes occurring in stars [43,44]. In turn, models should be able to reproduce the dependence of the s-process efficiency on stellar metallicity.

### 3.1. s-process in S Stars

As mentioned above, intrinsic S-type stars are among the first objects on the TP-AGB to show self-produced s-process elements in their photosphere. Several observational studies have investigated s-process elemental abundances in both extrinsic and intrinsic S stars, despite the difficulty of finding heavy element lines in spectra heavily blended with molecular (TiO and ZrO) bands. The landmark series of papers [34,45,46] and subsequent studies [19,20,32,47,48], together with s-process modelling investigations [4,5,49,50] have allowed to reach a good agreement between measured and predicted s-process abundances. Abundances of Sr, Y, Zr, Nb, Ba, Ce, Nd, Sm, Eu and even Tc can be measured, but, because of strong blends, with errors that might be in excess of 0.25 dex. The abundances in S-type stars are compatible with a s-process occurring radiatively in the interpulse region of low-mass stars. Evidences for the third dredge-up occurrence are now found in stars with masses as low as $\sim$1 $M_\odot$, as constrained by Gaia DR2 parallaxes [51].

### 3.2. s-process in C Stars

Since the pioneering work of [52,53], the s-process in SC stars has been shown [17] to be compatible with the S–SC–C evolutionary sequence. The luminosity of SC and N stars agrees well with AGB stars of masses 1.5–3 $M_\odot$. Galactic AGB carbon stars may reach $M_{bol} = -6$, implying that stars as massive as 5 $M_\odot$ can become carbon stars during their AGB phase [41].

### 3.3. Real-Time AGB Evolution

An exciting development concerning intrinsic AGB stars is the possibility to observe stellar evolution in real time. With a luminosity drop ($\sim$1% per year) and rapid decline of the pulsation period ($\sim$3.2 d/y) in the last 40 years [54], the O-rich, no-Tc Mira star T UMi is one of the best candidate stars to be at the start of a thermal pulse, though its observational changes could also be attributed to pulsation mode switching from the fundamental to the first overtone. However this mode switching could be induced by the thermal pulse itself [4]. Tc lines could be looked for in such stars to (in)validate the thermal pulse/dredge-up hypothesis.

## 4. "Cold Cases"

### 4.1. s-process in Extrinsic Stars

By observational selection bias, extrinsic stars are less variable than their more evolved intrinsic counterparts; consequently their abundances are easier to measure. Abundances have been derived in extrinsic S stars [19,35,55], barium stars e.g., [56–60], CH stars (e.g., [11,61,62]), CEMP-s and -sr stars ([11,63,64] and the SAGA database [65]), by order of decreasing metallicity.

These abundances globally validate the operation of the s-process in the radiative layers of the interpulse of AGB stars, adopting some fine-tuning, e.g., for the partial mixing of protons from the H-envelope into the C-rich region resulting from the previous He thermal pulse. Such a partial mixing is needed to create the $^{13}$C pocket necessary for the $^{13}$C($\alpha$,n)$^{16}$O neutron source to operate in low-mass stars [4,42,66,67]. Indeed, in low-mass giants, the $^{13}$C neutron source, operating at temperatures of about one hundred million kelvin, is favoured over the $^{22}$Ne($\alpha$,n)$^{25}$Mg neutron source (requiring temperatures in excess of $3.2 \times 10^8$ K), as proven by the radioactive pair $^{93}$Zr-$^{93}$Nb. Actually the $^{93}$Zr isotope is absent in the spectra of extrinsic stars because it has $\beta$-decayed into mono-isotopic $^{93}$Nb. Therefore, the Zr/Nb ratio measured today in extrinsic S stars is the same as the Zr/$^{93}$Zr abundance at the end of the TPAGB phase in their companion, provided the s-process contribution dominates over the initial heavy element abundances in the extrinsic star envelopes, which is the case in the most enriched extrinsic S stars. Fortunately, the Zr/$^{93}$Zr ratio is directly related to the neutron-capture cross sections of the various Zr isotopes. These neutron-capture cross sections, and thus the Zr/$^{93}$Zr ratio, depend on the s-process (interpulse) operation temperature, which can thus be constrained. An upper

temperatures of $\sim 2.5 \times 10^8$ K was inferred [55,60], favouring the $^{13}$C neutron source over the $^{22}$Ne one, which is indeed consistent with model expectations in low-mass AGB stars (1–3 $M_\odot$) [3,4].

The Zr-Nb pair not only acts as a thermometer but also, this time in intrinsic S stars, as a chronometer, just as the Tc-Ru pair, because $^{93}$Zr (resp., $^{99}$Tc) decays into $^{93}$Nb (resp., $^{99}$Ru) with $\tau_{1/2} = 1.53$ Myr (resp., $\tau_{1/2} = 0.21$ Myr). This allows to determine the time (1–3 Myr) since the onset of the s-process in intrinsic S-type stars, and this time lapse appears to be reasonably well correlated with other evolutionary indicators (like infrared excess) [55]. Because niobium abundance is too difficult to measure in carbon stars, this chronometer could not yet be applied to other TPAGB stars than S-type stars.

Finally, the Tc/Zr ratio is useful in that it allows to constrain the partial mixing parameter $\lambda_{\rm pm} = M_{\rm pm}/M_{\rm pulse}$, ratio between the mass of the region below the envelope where protons have been transported during the third dredge-up, and the pulse mass [68]. This way, $\lambda_{\rm pm}$ is constrained to values between 0.02 and 0.10.

We note that some bitrinsic stars have been uncovered, i.e., extrinsic (binary and Nb-rich) stars where the giant component has now reached its intrinsic (TPAGB and Tc-rich) stage [24]. Contrarily to extrinsic and intrinsic stars, these stars are thus both Tc- and Nb-rich.

As high-resolution abundance determinations became more and more frequent, the (weak) europium lines could be as easily measured as the more prominent barium lines, and stars with a mixed (s+r) abundance profile have been gradually uncovered at low metallicity (e.g., [63,69–71]): they are tagged CEMP-sr (or CEMP-r/s or CEMP r+s) stars. CEMP stars are metal-poor ([Fe/H] < −1) to very metal-poor ([Fe/H] < −2) stars with a carbon enrichment ([C/Fe] ≥ 1). Among those, the CEMP-s stars have [Ba/Fe] > 1, [Ba/Eu] > 0.5 or >0 and [Eu/Fe] < 1 while the CEMP-sr stars have as well [Ba/Fe] > 1 but [Eu/Fe] > 1 and 0 < [Ba/Eu] < 0.5. Finally, CEMP-r stars have [Ba/Eu] < 0 and [Eu/Fe] > 0.3 [63,70].

Both CEMP-s stars and CEMP-sr stars appear to belong to binary systems [11,72–74]. A careful classification is required to separate CEMP-s, -sr and -r stars [11,60]. The discovery of a star enriched in s- and r-elements at a higher metallicity ([Fe/H]$\sim$−0.7, [59]) puts a strong constraint on the range of metallicities at which such a mixed abundance profile may form: it may not be limited to very low metallicities. A similar abundance pattern has been found in the post-AGB star V4334 Sagittarii ([75], Sakurai's object).

Interestingly enough, [17] have noted that SC stars present a high Eu overabundance (0.5 to 1.0 dex), like the 2 barium stars HD 178717 and HD 121447. Whether (some) SC stars could represent the high-metallicity tail of the CEMP-sr objects still needs to be clarified.

The origin of CEMP-sr stars is still unclear. Single-event scenarios invoking the injection of protons in the He burning layers and leading to the so-called i-process [5] have been proposed. Several sites have been explored: (i) low-metallicity, low-mass thermally-pulsing AGB stars [22,77,78]; (ii) core He flash in low-metallicity RGB stars [79]; (iii) thermal pulse phase of intermediate-mass or massive AGB (super-AGB) stars of low metallicity (e.g., [80,81]); (iv) final thermal pulse (e.g., [82]); (v) rapidly-accreting white dwarf [83]; (vi) mixing events between H- and He-burning layers in massive (30 $M_\odot$), low-metallicity ([Fe/H] < −2) stars [84].

A double-event scenario has also been proposed to explain the peculiar abundance profile of the CEMP-sr star RAVE J094921.8-161722 [71], where thorium detection seems to demand pollution from a neutron-star merger, while the s-process pattern requires an independent AGB contribution. The rather weak Th signal is based on one single blended Th II line ($\lambda$ 4019.120 Å). Confirming thorium detection in these stars is very important but will be a hard task, since basically all optical thorium lines are heavily blended by CH or $C_2$ molecular lines in CEMP stars, which are by definition carbon-enriched.

*4.2. s-process in Post-AGB Stars*

The post-AGB phase, a transient ($10^2$ to $10^4$ yrs) phase during which the stars, having experienced a superwind episode (mass loss rate up to $10^{-4}$ $M_\odot$/yr) evolve at nearly constant luminosity to very hot effective temperatures (to $T_{eff} \sim 10{,}000$ K) before entering their planetary nebula phase (reached when $T_{eff} \geq 20{,}000$ K), is interesting because their hotter photospheres render their spectra devoid of the strong molecular lines characterizing AGB spectra, while they should still bear the cumulated signatures of the nucleosynthesis events that occurred during their entire evolution [85]. Lead abundances show a large dispersion at a given metallicity, which is not accounted for by models. The latter overestimate [Pb/hs] or [Pb/ls] ratios [6] for [Fe/H] < −0.7. A large range of neutron densities seems to be required to explain this dispersion. Interestingly enough, some (s-process enriched) post-AGB stars are suspected to have experienced a truncated AGB evolution due to binary interaction, for example the binary s-process-rich post-AGB star IRAS 17279-1119 [86], which is in a too small orbit to have hosted an AGB star, and which is a good barium star progenitor candidate. The impact of this truncation on surface abundances remains to be evaluated: no difference has yet been found between the abundance profiles of extrinsic stars (Ba stars and the truncated post-AGB mentioned above) and those of genuine thermally-pulsing AGB stars, except for Tc and Nb of course. However, the very last stages of AGB evolution could be characterized by different conditions (last thermal pulses occurring with a very thin hydrogen envelope [87]) inducing some changes in both the (post-)AGB photospheric composition, and in the ejected material polluting the future extrinsic star. In this respect, investigating the already mentioned Sakurai object (V4334 Sgr) and its siblings would be very useful, since indeed the abundance pattern (s+r) of V4334 Sgr seems to be distinct from the (s-process) pattern of intrinsic TPAGB stars. On the contrary, a C-depleted, N-enriched barium star has been observed surrounded by a spiral-shaped ejecta attributed to a very late thermal pulse [88]; it is enriched in barium but apparently not in Eu, ruling out a s+r abundance profile.

Incidentally, a whole population of RGB-truncated, dusty stars have been discovered in the Magellanic Clouds [89]. They might be reminiscent of the "disk-sources" [7] binary post-AGB objects, characterized by a near-infrared excess and frequently by a depletion pattern in refractory elements, that are both produced by their circumbinary dusty disk.

**5. Discussion**

Many important problems remain in our understanding of the details of the s-process. Let aside the issue (already discussed) concerning the site(s) of the i-process, a selection of these problems is discussed below.

First, an important discrepancy between observations and models concerns carbon and s-process element enrichments. While models predict a simultaneous increase of carbon and s-process elements, since they are all brought up to the stellar surface by third dredge-up events, the measured [s/C] ratio in many TPAGB objects is not reproduced by the models. For example, at the level of the s-process enrichment encountered in S-type stars, models produce a C/O ratio larger than unity, and predict that the star should thus appear as a carbon star and not as an S-type star. This problem is illustrated in Figure 2: while [s/Fe] values between 0.2 and 1.4 are encountered [24] in S-type stars (where C/O < 1 by definition), C/O is expected to exceed unity for [s/Fe] > 0.5 already, according to STAREVOL models [21,22] of different metallicities (M = 2 $M_\odot$, −0.5 < [Fe/H] < 0). The comparatively large measured s-process overabundances for a modest C enrichment is at tension with the models: the latter evidently dredge up too much carbon with respect to s-process elements, and it is not clear how to reconcile the measured and predicted carbon and s-process element abundances.

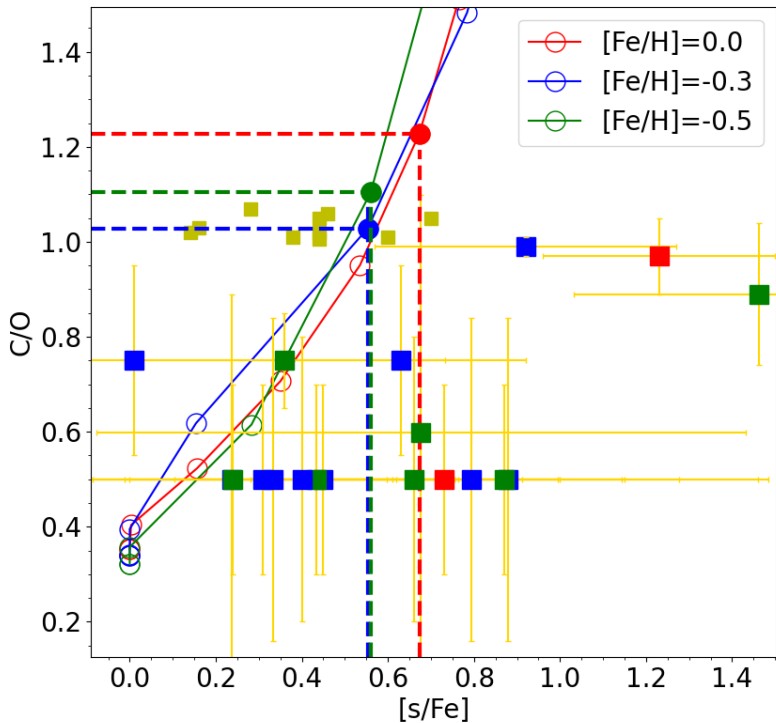

**Figure 2.** C/O ratio as a function of [s/Fe] for intrinsic (i.e., Tc-rich) S-type stars and carbon stars. Red, blue and green squares identify intrinsic S stars in the metallicity bins [0.0; −0.2], [−0.2; −0.4] and < −0.4. Tc-rich carbon stars from [90] are represented with small light green squares, clustering in the range C/O∼1–1.1. Predictions for a 2 M$_\odot$ STAREVOL model [21,22] at metallicities [Fe/H] = 0.0 (red), −0.3 (blue) and −0.5 (green) are overplotted. The empty circles along the tracks indicate the successive third dredge-up events along the TPAGB, while the three filled circles mark, for each considered metallicity, the third dredge-up events allowing to reach C/O > 1. According to models, no S-type star should be located at [s/Fe] larger than the ones indicated by the three blue, red and green filled circles (and corresponding vertical dashed lines). Figure adapted from [24].

Second, standard AGB models cannot explain some well-identified stellar families. The carbon stars of type J (representing ∼15% of all carbon stars) were already discussed. Extra-mixing acting during the He core flash or on the early AGB in low-mass stars has been mentioned to explain their peculiarities [39].

Maybe related are the R stars, differing from J stars because they are somewhat warmer and less luminous. R-type stars have been shown to be all single [91], suggesting that they may be the product of a merger event, since 20–30% of binaries are expected among such giants. With $\langle M_{bol} \rangle = -0.10 \pm 0.80$ mag [41], Gaia DR2 luminosities rule out the fact that they could be red clump stars, but rather place them on the upper part of the RGB. They might belong to an older, lower-mass population than the SC, N and J stars, and are probably not the progenitors of J-type stars, as previously suggested. The binarity of J-type stars has never been investigated. From a sample of a dozen J-type stars monitored with the Hermes spectrograph, no clear evidence of radial velocity variation that could be ascribed to binarity (rather than to photometric variability and pulsation) has been found (Van Eck and Jorissen, unpublished).

Third, most measured abundances of heavy elements in cool star spectra have been determined with the LTE hypothesis in 1D static model atmospheres. Checking the impact of using non-LTE computations for abundance determinations in 3-D model atmospheres would be very desirable. Non-LTE effects are suspected to be significant since abundances derived from neutral and ionized lines of some s-process elements do not agree. This problem has been detailed e.g., in [17,92], where in CY Cyg a difference of +0.25 dex (resp., 0.3 dex) is found between abundances of La (resp., Ce) derived from ionized and neutral

lines. The problem is very similar for Sr, Y and Zr, since the singly ionised species dominate for temperatures in the range 2500 K–5500 K, and non-thermal overionization in the upper atmosphere causes departure from LTE. When feasible, measuring high-excitation neutral and ionized lines, which form deeper in the atmosphere, could somewhat reduce the discrepancy [17]. Anyway it is extremely challenging to get abundance precision better than 0.2–0.3 dex in cool stars, because of (i) continuum placement uncertainties in highly depressed spectra, (ii) the dynamics of the atmosphere (stellar pulsations and shock waves), affecting (sometimes doubling) line profiles (iii) scatter due to oscillator-strength errors. The impact of 3D corrections is difficult to assess and might actually partially cancel the non-LTE ones in some specific cases.

## 6. Conclusions

Nucleosynthesis science is less than a century old and has been incredibly refined and tested since its foundation. The synthesis of heavy elements in stars, in particular, has reached a detailed level of understanding. As the abundance precision and accuracy improve, some tensions with models have appeared, revealing shortcomings in our treatment of physical processes as diverse as atomic transitions, convection or binary interaction, to name only a few. Observation of specific evolutionary phases, sometimes encountered only in binary systems where interaction occurred (in particular extrinsic stars), is able to bring additional constraints. The large surveys, compensating their sometimes less sophisticated analysis techniques by the strength of the statistics, will reveal new trends at the galactic and extra-galactic levels, provided that the distinction between extrinsic and intrinsic stars is properly assessed. A detailed understanding of TPAGB nucleosynthesis at different ages [93] and within galaxies seems now, more than ever, within reach.

**Author Contributions:** Conceptualization, S.V.E.; formal analysis, S.V.E.; investigation, S.V.E., S.S., L.S.; resources, S.S.; writing—original draft preparation, S.V.E.; writing—review and editing, S.V.E., L.S.; visualization, S.S. All authors have read and agreed to the published version of the manuscript.

**Funding:** This work has been supported by the Fonds de la Recherche Scientifique (FNRS, Belgium) and the Research Foundation Flanders (FWO, Belgium) under the EoS Project nr 30468642. SVE thanks Fondation ULB for its support.

**Institutional Review Board Statement:** Not applicable.

**Informed Consent Statement:** Not applicable.

**Acknowledgments:** We thank A. Jorissen for comments that helped us improve our manuscript.

**Conflicts of Interest:** The authors declare no conflict of interest. The funders had no role in the design of the study; in the collection, analyses, or interpretation of data; in the writing of the manuscript, or in the decision to publish the results.

## Notes

1　More precisely, $L_{3DU}$ is defined as the largest luminosity reached during the deepening of the convective envelope in the former pulse region.

2　BD +06 2063 , HD 96360, BD -13 4495, BD +16 3426, HD 189581, all no-Tc stars [34,35].

3　This specificity (namely C/O = 1) is often, erroneously, attributed to S stars in the literature, but S stars actually have $0.5 < C/O < 1$.

4　Whether a thermal pulse is a necessary and/or sufficient condition to induce a Mira pulsation mode switch is not presently known.

5　So-called "intermediate" process because it requires neutron densities ($n_n \sim 10^{12}$–$10^{15}$ cm$^{-3}$) intermediate [76] between those of the s-process ($n_n \sim 10^8$ cm$^{-3}$) and of the r-process ($n_n > 10^{24}$ cm$^{-3}$).

6　ls (resp., hs) denoting elements from the first (resp., second) heavy element peak.

7　as opposed to the "shell-source" post-AGB, the latter presenting a mid-infrared excess produced by the expanding spherical circumstellar envelope of the former AGB star [85].

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
