# Peer review of "Insights into AGB Nucleosynthesis Thanks to Spectroscopic Abundance Measurements in Intrinsic and Extrinsic Stars"

_universe, doi:10.3390/universe8040220_

Round 1
Reviewer 1 Report
The manuscript "Insights into AGB nucleosynthesis thanks to spectroscopic abundance measurements in intrinsic and extrinsic stars" by Van Eck & Shetye gives a good overview of the status of the knowledge of nucleosynthesis of heavy elements in AGB stars. This includes the successes and open questions regarding the measurement and interpretation of elemental abundances in stars which show effects of AGB nucleosynthesis.
The article reads well and has a good structure, clear explanations, and illustrative figures. It represents an important contribution to the field and reflects sufficiently high scientific standards to warrant its publication in Universe.
A few improvements and corrections should be made before publication, as marked in the attached annotated PDF file, and summarized below. I do not need to see the revised manuscript.
In general, the interpretation of a figure should only be in the main text, not in the figure caption. Parts of figure captions to be removed are marked.
A few additional references are needed, e.g. for MARCS model atmospheres, for the abundances of the S-type stars shown in Fig. 2, for the possible explanation of J-type carbon star abundances, and for the study of J-type stars with Hermes.
In the Abstract, some words about the s-process should be included, and the type of stars which the article focuses on should be mentioned (suggestion: "asymptotic giant branch stars, carbon-enhanced metal-poor stars, and post-AGB stars").
In the Introduction, a brief description of the s-process should be added.
In Sect. 2.1. "S-type stars", paragraph 1, a feature of Fig. 1 is described as "Two stars are however discrepant in Figure 1: HD 150922 and BD -10° 1977, both extrinsic stars in the [Fe/H]=-0.5 bin falling above L3DU."
However, in the figure the two stars have red symbols (open squares) and thus [Fe/H]=0.
Please clarify and correct this.
In two places, "Gaia eDR3" should be replaced by "Gaia DR2" to be consistent with the cited sources (in Sect. 3.1, line 151, and in Sect. 5, line 288).
In Sect. 4.1 some general information about CEMP stars and their metallicity range should be added.

Author Response
Please see below our point-by-point response to the referee.
The manuscript "Insights into AGB nucleosynthesis thanks to spectroscopic abundance measurements in intrinsic and extrinsic stars" by Van Eck & Shetye gives a good overview of the status of the knowledge of nucleosynthesis of heavy elements in AGB stars. This includes the successes and open questions regarding the measurement and interpretation of elemental abundances in stars which show effects of AGB nucleosynthesis.
The article reads well and has a good structure, clear explanations, and illustrative figures. It represents an important contribution to the field and reflects sufficiently high scientific standards to warrant its publication in Universe.
A few improvements and corrections should be made before publication, as marked in the attached annotated PDF file, and summarized below. I do not need to see the revised manuscript.
—>We checked the annotated PDF file and carefully took into account all its comments (changes are identified as bold text). Some further answers are added below.
In general, the interpretation of a figure should only be in the main text, not in the figure caption. Parts of figure captions to be removed are marked.
—>Corrected.
A few additional references are needed, e.g. for MARCS model atmospheres,
—>Added.
for the abundances of the S-type stars shown in Fig. 2,
—>Added.
for the possible explanation of J-type carbon star abundances,
—>Added.
and for the study of J-type stars with Hermes.
—>This is yet unpublished work, it is now mentioned as such.
In the Abstract, some words about the s-process should be included, and the type of stars which the article focuses on should be mentioned (suggestion: "asymptotic giant branch stars, carbon-enhanced metal-poor stars, and post-AGB stars").
—>Done.
In the Introduction, a brief description of the s-process should be added.
—>Done.
In Sect. 2.1. "S-type stars", paragraph 1, a feature of Fig. 1 is described as "Two stars are however discrepant in Figure 1: HD 150922 and BD -10° 1977, both extrinsic stars in the [Fe/H]=-0.5 bin falling above L3DU."
However, in the figure the two stars have red symbols (open squares) and thus [Fe/H]=0.
Please clarify and correct this.
—>The two red squares identified by the referee are actually not discrepant, because they fall to the left of the red dashed line, hence in the Tc-poor region (before the first third-dredge-up); it is therefore expected indeed that they are Tc-poor.
Actually, the discrepant stars fall in the [Fe/H]=-0.5 bin and are accordingly plotted as green squares. They are not easy to see because they fall in a crowded region; one falls on the top of an intrinsic (triangle) star. They fall to the right of the green dashed line, therefore in the Tc-rich region (after the first third dredge-up); they should thus be Tc-rich, but are not, this is why they are discrepant.
Some details have been added in the text to help the reader to identify properly the discrepant stars.
In two places, "Gaia eDR3" should be replaced by "Gaia DR2" to be consistent with the cited sources (in Sect. 3.1, line 151, and in Sect. 5, line 288).
—>We thank the referee for noting this. It has been checked and corrected everywhere.
In Sect. 4.1 some general information about CEMP stars and their metallicity range should be added.
—>Done
Reviewer 2 Report
Dear authors,
I found this review interesting in the intent of summarizing all the observational characteristics of different classes of stars as a function of the presence (or not) of s-process elements. I have some major comments with the intention of improving the manuscript and making the topic clearer also to readers not really close to the field:
1. An introduction on the s-process nucleosynthesis in AGB stars is missing. What is it and where/how it occurs? Which are the nuclear channels that can activate it and the conditions to be satisfied in order to produce s-process elements. Some words on the evolutionary conditions that characterize the stars experiencing the s-process nucleosynthesis can also introduce and clarify why we expect (or not) to see some elements in a determined range of masses. Some of these information are available along the text (e.g. in section 4.1), but summarizing them in the intro would make the paper clearer to the readers.
2. Why do not talk about s-process in general before discussing the Tc cases? I suggest the authors to discuss the s-process cases (section 3) before the specific case of Tc (section 2).
3. Section 2.1: please introduce the models with which the data have been compared (not only in the caption of Fig.1).
4. Section 2.1: please clarify how it has been verified that no-Tc stars are extrinsic: on the basis of their luminosities (which are too low to be AGB stars) or are there other evidences?
5. Fig. 1: some aspects of this figure have to be clarified:
- please, indicate the initial masses along the theoretical lines. There are many up and down in the lines, it seems strange that each point represents a different mass model. Please explain.
- how is defined the L_{3du}? Because the lowest L_{3du} is around 2000 L_sun which is comparable with the luminosity at the Tip of the RGB. An efficient TDU is experienced by AGB stars when the luminosity is above ~4000 Lsun.
- please, add the reference of the sample in the caption of the figure.
- among the various evolutionary models available in the literature which include the s-process nucleosynthesis during the AGB phase, why the STAREVOL ones have been chosen? Is there a specific reason?
6. section 3.1: please, add some words on which are the s-process abundances expected and observed in this class of stars.
7. section 3.3: in this section it is described how Tc could be used as an evolutionary constrain. Why is it not included in section 2 together with the rest of the discussion regarding Tc? I suggest to include it in that section.
Author Response
—> We reply point-by-point to the referee below.
Dear authors,
I found this review interesting in the intent of summarizing all the observational characteristics of different classes of stars as a function of the presence (or not) of s-process elements. I have some major comments with the intention of improving the manuscript and making the topic clearer also to readers not really close to the field:
- An introduction on the s-process nucleosynthesis in AGB stars is missing. What is it and where/how it occurs? Which are the nuclear channels that can activate it and the conditions to be satisfied in order to produce s-process elements. Some words on the evolutionary conditions that characterize the stars experiencing the s-process nucleosynthesis can also introduce and clarify why we expect (or not) to see some elements in a determined range of masses. Some of these information are available along the text (e.g. in section 4.1), but summarizing them in the intro would make the paper clearer to the readers.
—>Now done in the Introduction.
- Why do not talk about s-process in general before discussing the Tc cases? I suggest the authors to discuss the s-process cases (section 3) before the specific case of Tc (section 2).
—>We have chosen to discuss Tc detection in the first place to stress the fact that it is the observational diagnostic to identify TPAGB stars. We then discuss in Sect 2 in which stars Tc has been effectively detected, and this allows us to describe various types of TPAGB stars. A discussion of s-process in TPAGB stars then follows in Sect 3. We agree with the referee that other logical links would be possible, but this would need to assume a-priori which stars are TPAGB stars, while from an observational point of view, we do not know this, except if Tc content is known. We would then prefer to keep this order. We have clarified the adopted plan in the Introduction.
- Section 2.1: please introduce the models with which the data have been compared (not only in the caption of Fig.1).
—>Done.
- Section 2.1: please clarify how it has been verified that no-Tc stars are extrinsic: on the basis of their luminosities (which are too low to be AGB stars) or are there other evidences?
—>This is explained by the sentence: “As required by the extrinsic scenario, no-Tc stars have been found to be binaries [27–30]”. It is thus through radial velocity monitoring that no-Tc stars have been checked to be extrinsic. However we agree with the referee that the logic was not very clear in the section, since there was a mix between “no-Tc” and “extrinsic”. We have clarified this, keeping only the observational diagnostic “no-Tc” in the first place, and then mentioning “extrinsic” stars once their binarity is mentioned.
- Fig. 1: some aspects of this figure have to be clarified:
- please, indicate the initial masses along the theoretical lines.
—>The figure will get very crowded if all masses are indicated, but we did follow the referee suggestion and indicated some masses. We now refer to Fig. 4 of Shetye et al. 2021 for more details.
There are many up and down in the lines, it seems strange that each point represents a different mass model. Please explain.
—>Indeed many models are considered of various masses, separated by a 0.1Msun step between 1.3 and 3Msun, then increasing slightly (0.2 or 0.3 mass step) till 5Msun. So each angular point in the 3DU line indeed represents a different mass model.
How is defined the L_{3du}? Because the lowest L_{3du} is around 2000 L_sun which is comparable with the luminosity at the Tip of the RGB. An efficient TDU is experienced by AGB stars when the luminosity is above ~4000 Lsun.
—>The luminosity varies during the third dredge-up event. The adopted L_{3du} corresponds to the largest luminosity reached during the deepening of the envelope in the former pulse region. This is now described in a footnote. From mass track to mass track this luminosity may fluctuate because the structure is readjusting in radius and the chemistry is changing rapidly (carbon abundance impacting the opacities). More precisely, the start of the 3DU is identified as soon as the mass coordinates at the basis of the convective envelope is smaller than the mass coordinate at the top of the pulse. Thus between the maximum extension of the pulse and the start of the 3DU, the radius and thus the luminosity can significantly change. Depending on the stellar mass and the pulse intensity, it can take different amount of time, and this will result in luminosity fluctuation.
- please, add the reference of the sample in the caption of the figure.
—>Added.
- among the various evolutionary models available in the literature which include the s-process nucleosynthesis during the AGB phase, why the STAREVOL ones have been chosen? Is there a specific reason?
—>The STAREVOL models have been chosen in the whole serie of papers on S stars, because they include up-to-date reaction rates, couple the full s-process network with the transport equation and is available locally for production of tailored models (Goriely & Siess, A&A 2018, 609, A29; Choplin et al., A&A 2021,648, A119).
They also compare well with other models (see e.g. comparison, concerning s-process elements, between Fruity, MSE and STAREVOL models in Fig. 14 of De Smedt et al. (2015, A&A 583, A56).
- section 3.1: please, add some words on which are the s-process abundances expected and observed in this class of stars.
—>All s-process elements are expected to be over-abundant in S stars, since these stars have produced themselves (or have been polluted) by products of s-process nucleosynthesis. We understood that the referee was lacking some details on the elements strongly produced by the s-process, and that can be effectively observed in such stars, given their crowded spectra; we have now added explanatory sentences and we list these chemical elements in Sect 3.1.
- section 3.3: in this section it is described how Tc could be used as an evolutionary constrain. Why is it not included in section 2 together with the rest of the discussion regarding Tc? I suggest to include it in that section.
—>Section 2 is an overview of Tc effective detections in stars. In Sect 3.3, we just mention a prospective idea that has not yet been implemented, namely to measure Tc in stars being at the start of a thermal pulse. Since, to our knowledge, this has never been done, we think it is clearer to keep this observational suggestion in the paragraph where we mention the possible targets where Tc detection could be attempted.
Reviewer 3 Report
In this paper, the astronomical observation of low and middle mass stars is reviewed to discuss stellar nucleosynthesis, in particular slow neutron capture reaction process (s-process). The s-process is one of the major nucleosynthesis for generating of the elements heavier than the iron group. The correlation between the observed elemental abundances and the class of stars is discussed. Tc is one of key elements for the study of the s-process, because there are no stable isotopes in the Tc element and the observed Tc is freshly synthesized in a star. Thus, Tc stars are discussed. The AGB star is considered to be the dominant source of the s-nuclei. The correlation between the phase of evolutional states of the AGB stars and the observed spectra are also discussed. The review for these starts and their spectra is useful for wide audiences having an interest for the s-process and its astronomical observation. In discussion part, the unresolved problems are discussed. This is also useful. The authors well describe these topics except for the i-process. In conclusion, I recommend its publication in the journal, Universe, with following minor changes.
1. Various stars are discussed. Most stars for low or middle mass stars are well explained, but the only abbreviated notation for extremely metal poor stars such as CRMP-s is presented. It is not kind for audiences. I suggest to add their brief explanation.
2. The fact that Tc has no stable isotopes should be written more clearly.
Author Response
—> Please find our point-by-point response to the referee below.
In this paper, the astronomical observation of low and middle mass stars is reviewed to discuss stellar nucleosynthesis, in particular slow neutron capture reaction process (s-process). The s-process is one of the major nucleosynthesis for generating of the elements heavier than the iron group. The correlation between the observed elemental abundances and the class of stars is discussed. Tc is one of key elements for the study of the s-process, because there are no stable isotopes in the Tc element and the observed Tc is freshly synthesized in a star. Thus, Tc stars are discussed. The AGB star is considered to be the dominant source of the s-nuclei. The correlation between the phase of evolutional states of the AGB stars and the observed spectra are also discussed. The review for these starts and their spectra is useful for wide audiences having an interest for the s-process and its astronomical observation. In discussion part, the unresolved problems are discussed. This is also useful. The authors well describe these topics except for the i-process. In conclusion, I recommend its publication in the journal, Universe, with following minor changes.
- Various stars are discussed. Most stars for low or middle mass stars are well explained, but the only abbreviated notation for extremely metal poor stars such as CRMP-s is presented. It is not kind for audiences. I suggest to add their brief explanation.
—> Now better explained both in the Introduction and in Section 4.
- The fact that Tc has no stable isotopes should be written more clearly.
—>Indeed, this is now clearly mentioned (beginning of Sect. 2).
Round 2
Reviewer 2 Report
Dear authors,
I am fine with your replies and text added to the manuscript in response to my warnings and suggestions. Please check the citations, many of them appear as question marks.
Kind regards
Author Response
Reference problem has been fixed and some sentences have been clarified.
